# Revisiting the Old Data of Heat Shock Protein 27 Expression in Squamous Cell Carcinoma: Enigmatic HSP27, More Than Heat Shock

**DOI:** 10.3390/cells11101665

**Published:** 2022-05-17

**Authors:** Shutao Zheng, Yan Liang, Lu Li, Yiyi Tan, Qing Liu, Tao Liu, Xiaomei Lu

**Affiliations:** 1State Key Laboratory of Pathogenesis, Prevention and Treatment of High Incidence Diseases in Central Asia, Clinical Medical Research Institute, The First Affiliated Hospital of Xinjiang Medical University, Urumqi 830054, China; zhengshutao@xjmu.edu.cn (S.Z.); tyy515138@163.com (Y.T.); liuqing@xjmu.edu.cn (Q.L.); 2Department of Pathology, Basic Medicine College, Xinjiang Medical University, Urumqi 830017, China; vayee0809@163.com; 3Department of Clinical Laboratory, First Affiliated Hospital of Xinjiang Medical University, Urumqi 830054, China; LuLi051212@163.com (L.L.); lt02-16@163.com (T.L.)

**Keywords:** heat shock protein 27(HSP27), squamous cell carcinoma (SCC), clinicopathological, prognostic

## Abstract

Initially discovered to be induced by heat shock, heat shock protein 27 (HSP27, also called HSPB1), a member of the small HSP family, can help cells better withstand or avoid heat shock damage. After years of studies, HSP27 was gradually found to be extensively engaged in various physiological or pathophysiological activities. Herein, revisiting the previously published data concerning HSP27, we conducted a critical review of the literature regarding its role in squamous cell carcinoma (SCC) from the perspective of clinicopathological and prognostic significance, excluding studies conducted on adenocarcinoma, which is very different from SCC, to understand the enigmatic role of HSP27 in the tumorigenesis of SCC, including normal mucosa, dysplasia, intraepithelial neoplasm, carcinoma in situ and invasive SCC.

## 1. Introduction

Cancer cells are characterized by multiple molecular alterations. Nevertheless, making an overview of the major proteins involved in oncogenic signaling pathways seems to be impractical for the time being. In this scenario, HSP27 is among the altered proteins in cancer cells. To date, the key roles HSP27 plays in the biology of cancer have been gradually acknowledged, including their interplay with other proteins that regulate cancer development via their activities implicated in apoptosis, mitotic signaling pathways, epithelial-to-mesenchymal transition and metastatic processes. In recent decades, HSP27 was found to participate in the pathogenesis of squamous cell carcinoma (SCC), with a large number of published studies reporting on the involvement of HSP27 in tumors. However, in this critical review, we excluded all of the studies unrelated to SCC. In addition, some literature concerning basic and mechanistic studies on HSP27 but not performed in the setting of SCC was also not considered. For this critical review, we only focused on the clinicopathological meaning of HSP27 expression in the context of SCC, irrespective of its biological roles and biochemical mechanisms proposed in vitro in cancer cell lines and in vivo in a cancer animal model.

## 2. Patterns of HSP27 Expression in SCC

Either reduced or elevated expression of HSP27 was the first indication of the putative role of HSP27 in SCC. Some early studies analyzed the expression pattern of HSP27 in SCC. For instance, one of the first reports analyzing the relationship between HSP27 expression and SCC was a study of skin squamous cell carcinoma (SSCC) [1], demonstrating the weak expression of HSP27 in SSCC compared with normal skin. Likewise, in another study of the human epidermis and its corresponding malignancy, HSP27 expression was hardly detectable in SSCC [2]. In contrast, HSP27 can be detected in normal epidermis, suggesting that a loss of HSP27 might be a marker of epidermal malignancy.

However, it is quite a different scenario in esophageal squamous cell carcinoma (ESCC). In contrast to the lack of HSP 27 in SSCC observed in the literature, a couple of studies [3,4], including our own performed in Kazakh’s ESCC tissues [5,6], all found that HSP27 was over-expressed in ESCC relative to normal controls. However, this notion was challenged by several contrary findings from ESCC [7,8], where HSP27 was reported to be significantly reduced in ESCC compared with matched normal controls. Along the same line, in head and neck squamous cell carcinoma (HNSCC) and oral squamous cell carcinoma (OSCC), the expression status of HSP27 remains controversial, with some [9,10] reporting that HSP27 was up-regulated and others [11,12] reporting it was down-regulated in neoplastic tissues relative to normal controls. The chief reasons accounting for these differences are largely unclear, but they may be due to some issues from technical aspects, such as the different primary antibodies used, interobserver variation when immunoscoring and even different cancerous tissues with hardly any comparable TNM classifications.

The two articles [1,2] reviewed above show that a lack of HSP27 might be commonly seen in SSCC. The biochemical roles of HSP27 in SSCC are still unclear in the absence of additional evidence. However, a mechanistic study revealed that HSP27 mediated thermotolerance but did not protect keratinocytes from UV-induced cell death [13]. In this report, Trautinger F et al. [13], using a transfection vector harboring the human gene for HSP27, demonstrated that the over-expression of hsp27 confers increased resistance to hyperthermia, as opposed to hydrogen peroxide-mediated oxidative injury, in the skin squamous cell carcinoma cell line A431. The examples shown above clearly indicate that HSP27 can be anti-tumorigenic or pro-tumorigenic in SCC of different types. Despite this, the role of HSP27 in SSCC and ESCC is largely unknown, and it remains to be illuminated in subsequent studies. The patterns of HSP27 expression in the pathogenesis of SCC are tabulated in Table 1.

## 3. Clinicopathological Meaning

The first report on the clinicopathological significance of HSP27 expression was from oral squamous cell carcinoma (OSCC) by Mese H et al. [19], reporting that no correlation was observed between HSP27 expression and the clinical stage, lymph node metastasis or histological grade of OSCC tissues, employing a total of 40 cases. Consistent with this finding, in another study regarding OSCC by Lo Muzio L et al. [11], the authors did not find a relationship between HSP27 expression, gender or tumor size after statistical analysis in their study involving 79 cases of OSCC and 10 cases of normal mucosa as controls. Two years later, Lo Muzio L and colleagues [12] retrospectively analyzed the prognostic value of HSP27 expression in OSCC with 57 clinical samples. No significant association was identified between HSP27 expression and the age, sex, histological grade, TNM stage or clinical stage of OSCC. After combining the three studies reviewed above concerning OSCC, it seemed to be tentatively concluded that HSP27 was not associated with clinicopathological parameters.

Inconsistent with this tentative conclusion, a recent study contributed by Karri RL et al. [20] was observed, employing 30 samples of epithelial dysplasia (10 mild dysplasia, 10 moderate dysplasia and 10 severe dysplasia/carcinoma in situ cases), showing a significant correlation between HSP27 expression and the severity of dysplasia and well-differentiated OSCC.

Aside from OSCC, what about the clinicopathological significance of HSP27 expression in other kinds of SCC? Driven by this question, we searched for and retrieved relevant literature related to the theme we framed at the outset. In another study performed on tongue squamous cell carcinoma (TSCC), Mohtasham N et al. [21] used immunohistochemical staining to detect the expression of HSP27 and HSP105. In this study, enrolling 56 cases of clinical samples covering the I to III histopathological grades of TSCC, an inversely significant relationship was found between the expression of HSP27 and the grade of the disease. Moreover, there was also an inversely significant correlation between early-stage and advanced-stage in terms of HSP27 expression. Regrettably, the authors failed to analyze the prognostic meaning of HSP27 in their study.

In another study about TSCC involving 15 normal tongue mucosa, 31 dysplastic lesions, 80 primary TSCC and 32 lymph node metastases, Wang A et al. [17] reported, using immunohistochemistry, that reduced HSP27 expression was remarkably associated with poor differentiation in primary TSCC tissues. Specifically, the expression of HSP27 appeared to gradually decrease with the development of TSCC. The data reviewed above explicitly suggested that HSP27 seemed to have important implications in the differentiation and progression of TSCC, although the implications are unclear.

In another study conducted in ESCC dealing with 28 cases of ESCC clinical samples, using an immunohistochemistry approach, Luz CC and colleagues [22] appraised the expression of HSP27 and HSP 70 at the same time, discouragingly observing that neither HSP27 nor HSP70 was markedly associated with gender, age, surgical margin, lymph node metastasis or tumor differentiation. Noticeably, the major limitation from which this study suffered was that the sample size was rather limited, only 28 cases, leading to statistical powerlessness when stratifying for clinicopathological significance analysis. However, similar observations were made by Zhang Y et al. [23] regarding HSP27 expression in ESCC. In this study, enrolling 162 cases, mainly adopting immunostaining, the authors did not find any significant correlation between HSP27 expression and clinicopathological information, including age, gender, differentiation, tumor size and depth of infiltration. However, a significant correlation was observed between HSP27 expression and lymph node metastasis. Regarding survival analysis, no attempt was ever made to determine whether the expression of HSP27 is associated with prognosis. All of these findings are summarized in Table 2. Taken together, through combing through the literature analyzed above, no consensus seems to be reached on the prognostic significance of HSP27 expression in patients with squamous cell carcinoma. The correlation between the expression of HSP27 and clinicopathological parameters is still controversial and requires investigation with a larger sample size.

## 4. Prognostic Significance

As stated in the preceding section, generally, a small sample size was a striking limitation in earlier studies exploring the prognostic and clinicopathological significance of HSP27 expression in SCC. Under this circumstance, correlational analysis was notpossible when undergoing further stratification. For example, one of the first investigations describing the relationship between HSP27 expression and SCC was a study performed in TSCC involving 24 cases [14]. In this investigation, HSP27 expression was shown to be negative in the normal epithelium of the tongue, while in dysplastic lesions, HSP27 was positive. In squamous cell carcinoma, positive immunostaining of HSP27 is commonly seen in the cytoplasm of suprabasal tumor cells. In view of the limited sample size involved, the authors did not analyze the correlation between HSP27 expression and clinical stage, lymph node metastasis or histological grade. Prognostically, these authors claimed that no significant correlation was found between HSP27 and the survival of TSCC, without any corresponding data provided.

In another study dealing with 50 cases of HNSCC tissue samples by means of immunohistochemistry, HSP27 was demonstrated to be positively expressed in normal upper respiratory tract squamous mucosa and in tumors with T1 and T2 stage, with its pathological significance still being unclear. In this study, HSP27 did not correlate with 5-year survival [25]. Using 77 cases of ESCC tissue specimens, Shiozaki H et al. analyzed the prognostic significance of HSP27 expression in ESCC, showing that the abnormal expression of HSP27 was pronouncedly associated with shorter survival. In the study by Shiozaki H et al. reduction or loss was defined as the abnormal expression of HSP27. Similarly, in an OSCC study enrolling 40 samples, the expression of HSP27 was found to be inversely correlated with survival. Consistently, Lo Muzio L et al. [11] concluded, using 79 OSCC tissues and 10 normal mucosa samples as controls, that patients with negative/reduced HSP27 expression had significantly lower survival rates than those with positive HSP27 expression in OSCC. Subsequently, a retrospective analysis of the prognostic value of HSP27 in OSCC was conducted by Lo Muzio L et al. [12], who utilized 57 cases, confirming that reduced HSP27 expression was an independent significant prognostic predictor for inferior overall survival by Cox regression analyses.

The results described above may help us shape an opinion that decreased or reduced HSP27 expression can predict the poor survival of patients with OSCC. The same is true in TSCC. Attempting to further evaluate the prognostic value of HSP27 expression, Wang A et al. [17] examined the expression of HSP27 by immunohistochemistry in a total of 158 clinical specimens, including 15 cases of normal tongue mucosa, 31 cases of dysplastic lesions, 80 cases of TSCC and 32 lymph node metastases, demonstrating that lower Hsp27 expression was markedly correlated with worse overall survival. Through multivariate analyses, HSP27 was shown to be an independent significant predictor of survival in TSCC. In line with these findings, Lomnytska MI et al. [29] assessed the diagnostic value of HSP27 in cervical squamous cell carcinoma (CSCC), enrolling a total of 66 clinical samples, including 8 normal cervix, 13 cervical intraepithelial dysplasia and 45 cases of CSCC with stages from IA1 to III, confirming that reduced HSP27 was markedly associated with worse relapse-free and overall survival. A similar conclusion was achieved in a meta-analysis performed by Wang XW and colleagues [30] in ESCC, with nine studies and 801 cases included. Their results supported the proposal that reduced HSP27 expression was associated with the poor survival of ESCC. However, this notion was somewhat challenged by the investigation conducted by Luz CC et al. [22] regarding HSP27 expression in ESCC, showing that no relationship can be observed between HSP27 expression and survival.

The studies reviewed above established an association between HSP27 expression and SCC survival based on the cytoplasmic localization of HSP27. The prognostic value of the nuclear sublocalization of HSP27 was seldom described, except for one study performed by Kaigorodova EV et al. [31], who originally reported in laryngeal squamous cell carcinoma (LSCC), totaling 50 cases, that the nuclear expression of Hsp27, whether phosphorylated or unphosphorylated, was a molecular marker of an unfavorable prognosis. This observation suggests that the nuclear expression of HSP27 could predict the outcome of LSCC, but this will need to be confirmed in subsequent studies in the setting of SCC, with a larger sample size.

## 5. Participation in Proliferation

Proliferation, one of the hallmarks of cancer cells, involves many oncogenes that interact and cooperate. Previously, the influence exerted by HSP27 on tumor growth both in vivo and in vitro was evaluated by Kindas-Mugge I et al. [32] using the melanoma cell lines A375 and A431, which are epidermal squamous carcinoma cell lines that are commonly used in the laboratory and in nude mice.

When an HSP27 expression vector was introduced into A375 and A431 cells, a lower proliferation rate was observed relative to the control. Extended to in vivo, a distinct delay in tumor development was observed in nude mice burdened with cells over-expressing HSP27. Surprisingly and unexpectedly, this initial delay observed at the outset of the study became increasingly indistinguishable in the growth dynamics in some of these animals compared to the control, with no difference eventually being observable. These results were basically in line with subsequent data provided by Zhu Z et al. [33], who similarly found no significant causal relationship between HSP27 and HNSCC cell proliferation. Consistent with these two articles, a statistically indistinguishable change in the proliferation rate of HNSCC cells was observed before and after HSP27 was silenced, as assayed by MTS with lentivirus-based short hairpin RNA (shRNA) plasmid, in a recent study by Zhu ZK and colleagues [34] in HNSCC. These findings explicitly indicate that the expression of HSP27 is not closely associated with the proliferation of SCC cells.

What is surprising is that the observations made in our own study in ESCC did not support these previous studies in terms of proliferation. In our own study [6], knockdown of HSP27 using a small interfering RNA (siRNA) technique in KYSE150 and KYSE450 ESCC cells retarded the growth of ESCC cells. The possible reasons leading to the discrepancy are largely unknown, partly because of the different techniques used. In addition, one study [35] that should be mentioned here is that HSP27 was identified to be an interacting protein for LINC00551, a kind of long non-coding RNA that was revealed to be significantly down-regulated in ESCC. Mechanistically, decreased phosphorylation of HSP27 mediated by LINC00551 can retard proliferation and suppress the invasion of ESCC cells.

## 6. Involvement in Metastases

In addition to participating in proliferation, HSP27 also has some roles in mediating the metastasis of SCC. To be exact, when analyzing the previous studies revolving around HSP27 that were conducted using two-dimensional SCC cell lines in vitro, it is not so much metastasis as motility, including migration and invasion, that is affected by HSP27. These earlier studies did not involve an in vivo animal model to investigate the role of HSP27 in mediating the metastasis of SCC.

For example, in ESCC, the first disputable conclusion was made by Xue L et al. [7] that HSP27 was involved in the migration and invasion of EC9706 and Eca109 cells, two frequently used ESCC cell strains derived from Chinese patients. In this study, the authors concluded that HSP27 could profoundly suppress the migration and invasion, which could both be referred to as mobility, of EC9706 and Eca109 cells based on wound-healing and Transwell experiments. However, these results seem to have been increasingly thrown into question by subsequent similar reports with conclusions contrary to those reported by Xue L et al. [7]. In our own report [6], using the ESCC cell lines KYSE-150 and KYSE-450, we found that HSP27 could remarkably promote the motility of ESCC cells. A similar conclusion that HSP27 can promote motility was found for types of cancer other than ESCC, for instance, HNSCC [33], supporting that HSP27 is a metastasis-promoting gene in SCC. Despite this, little was learned about how HSP27 promotes motility in ESCC cells. In contrast, a couple of studies with indisputable evidence [6,36] are available regarding HSP27 modulating epithelial-mesenchymal transitions of SCC.

## 7. Engagement in Chemoresistance

Aside from metastasis, chemoresistance is another leading reason for the recurrence of SCC. Unfortunately, what we know about HSP27 being engaged in chemoresistance primarily comes from experimental investigations carried out on adenocarcinoma rather than SCC. In contrast, very few studies explored the role of HSP27 in the chemoresistance of SCC. Recently, the engagement of HSP27 in the chemoresistance of ESCC was observed, with only one piece of literature describing that HSP27 is involved in chemoresistance in ESCC [37]. In their elegantly designed work presented by Liu CC et al. [37], the chemoresistance of esophageal cancer stem cells was found to be critically dependent on the activation of HSP27, which implies that Hsp27 could be a novel therapeutic target for treating esophageal cancer. However, the recognition of HSP27 involvement in chemoresistance in ESCC was rather limited and remains to be further verified.

In contrast, a flurry of research was conducted on other squamous cell carcinomas in reverse chronological order, including tongue [38], oral [39] and head and neck [40], all of which appeared to explicitly support the proposal that elevated HSP27 was heavily involved in chemoresistance of SCC, suggesting a possible causal relationship between elevated HSP27 and chemoresistance. However, this is not necessarily the case. This concept seemed to be challenged by direct evidence provided by Nakata B et al. [41] in vitro in HNSCC cell lines artificially made resistant to cisplatin, demonstrating that the emergence of cisplatin resistance did not lead to increases of HSP27 at the mRNA level. Instead, resistance to cisplatin seems to be more likely related to HSP27 inhibition at the mRNA level in HNSCC cell lines.

In support of this, Miyazaki T et al. [27], enrolling 61 patients with ESCC before treatment with chemoradiotherapy (31 patients) or radiotherapy (30 patients) and investigated the potential association of HSP27 expression with chemo or radiotherapy, revealing that HSP27 was the most reliable predictor of the effect of chemoradiotherapy and radiotherapy by multivariate analyses. Although several mechanisms [37,38,40] by which HSP27 is involved in chemoresistance were put forward in the context of SCC, the underlying mechanisms were not fully elucidated, with only an association or correlational relationship identified between up-regulated HSP27 and chemoresistance [42] in outline. Additionally, many biochemical details were missing that are left to be filled in. Although direct evidence established a causal relationship between elevated HSP27 expression and chemoresistance in SCC and adenocarcinoma settings [38,42,43], revealing that HSP27 is an important player in chemoresistance, mechanistically, many important biochemical details remain unclear and remain to be developed and tested.

However, a consensus was made by several studies [44,45,46,47] conducted in adenocarcinoma that chemoresistance was heavily dependent on the phosphorylation of HSP27. The underlying mechanism by which activated HSP27 results in chemoresistance in cancer cells remains undetermined. To better understand how HSP27 is involved in the chemoresistance of cancer cells, we drew a schematic diagram illustrating the mechanisms of HSP27 in chemoresistance (Figure 1). In this picture, the mechanism by which HSP27 participates in chemoresistance is shown; however, much detailed information is missing and needs to be filled in.

## 8. Engagement in Radioresistance

Similar to chemoresistance, radioresistance is another problem commonly encountered in SCC in the clinic. However, the underlying molecular mechanism can be distinctively different between chemo- and radioresistance even for the same molecule, such as HSP27. In HNSCC, attempting to uncover differences in radiation handling strategies, Muschter D and colleagues [48] analyzed the intracellular expression of HSP27 and HSP70 using an enzyme-linked immunosorbent assay (ELISA) approach in HNSCC cell lines subjected to different doses of radiation. Using normal human dermal fibroblasts and human dermal microvascular endothelial cells as controls, they showed that HSP27 was higher in HNSCC cells than in control cells and that the proportion of phosphorylated HSP27 was increasingly elevated with an increasing irradiation dose. The same was true for HSP70, which was detected in parallel with HSP27. These data suggest that HNSCC cells may take advantage of HSP27 to evade radiation-induced apoptosis.

Further supporting this suggestion, direct evidence from an in vivo mouse model and an in vitro cell culture investigation made by Hadchity E et al. [49] demonstrated that silencing HSP27 with OGX-427, a second-generation antisense oligonucleotide, was able to sensitize radiosensitive HNSCC cells to radiation. Nonetheless, two divergent and even conflicting reports also emerged. In an early Chinese study performed with nasopharyngeal carcinoma (NPC) [50], both HSP27 and HSP70 were simultaneously detected with immunohistochemistry in 58 NPC sample tissues, and no significant differences in HSP27 expression were identified between NPC with and without local residual lesions. Similarly, HSP27 expression was evaluated using immunohistochemistry in HNSCC treated by radiotherapy [40], underscoring that the over-expression of HSP27, whether in vivo or in vitro, has no conspicuous effect on the survival of tumors exposed to radiation compared with the control. Taken together, all of the evidence gleaned above explicitly indicates that the role HSP27 plays in the radioresistance of SCC remains largely controversial. To date, no study has been conducted to reconcile the conflicting results reported above.

## 9. Protection against Oxidative Stress

Apart from engagement in chemo- and radioresistance, HSP27 was also discovered to take part in cytoprotection from apoptosis induced by oxidative stress [51]. In regard to protection from oxidative stress, there remains some disagreement over whether it is phosphorylated or unphosphorylated HSP27 that plays a protective role against cell death and apoptosis mainly caused by oxidative stress, with some [52,53] supporting that HSP27 exerts a protective role, whereas others [54,55] argue that only phosphorylated HSP27 protects against oxidative stress.

Accumulating evidence [56] has shown that phosphorylation of HSP27 can be stimulated by a variety of stresses in addition to heat shock or hyperthermia, for example, oxidative stress or exposure to cytotoxic drugs (heavy metals, chemotherapy, alcohol). It can be imagined that when in such an environment, cells have to establish a protective mechanism to survive the adverse conditions and maintain their functions. Oxidative stress, generally referred to as excess production of reactive oxygen species (ROS) relative to antioxidant defense, can cause large amounts of ROS formation within cells, which in turn further amplifies oxidative stress, eventually leading to cell death and apoptosis. A great deal of experimental data indicated that HSP27 has a protective effect against oxidative stress owing to its ability to modulate intracellular redox homeostasis. Intracellular large aggregates of HSP27 are shown to be capable of regulating ROS production and therefore generating cytoprotection against oxidative stress [57]. Nevertheless, this activity can be somewhat mitigated by phosphorylation of HSP27, mainly through dissociating the HSP27 complexes into tetramers [58]. Consistently, one study contributed by Huot J et al. [59] revealed that over-expression of the wild-type, as opposed to nonphosphorylated HSP27 from humans, led to increased resistance against oxidative stress and enhanced survival after treatment with H_2_O_2_. Basically, in line with this report, another study [60] held a similar opinion that phosphorylated HSP27 seems to be dispensable for the protection of HSP27 against H_2_O_2_-induced oxidative stress. These data strongly suggest that HSP27 exerts cytoprotection against oxidative stress via different mechanisms, such as interactions with client proteins [61], prevention of protein aggregation and protection of the cytoskeleton. Considering the close relationship [62,63,64] between the development of SCC and oxidative stress, the studies mentioned above may help account for the phenomenon that high expression of HSP27 is frequently present in tumorous tissues relative to paired normal controls.

## 10. Participation in Epithelial-Mesenchymal Transition (EMT)

In addition to engagement with chemo- and radiotherapy, HSP27 was shown to take part in the epithelial–mesenchymal transition (EMT) of squamous cell carcinoma. Contrary to the conventional wisdom holding that EMT is a key event promoting metastasis in cancer [65], two separate seminal studies [66,67] published in the journal *Nature* challenged the entrenched opinion, demonstrating that EMT was dispensable for metastasis but seemed to be more implicated in chemoresistance of cancer cells. Given the controversy regarding metastasis, let us ignore it here. The role EMT plays in metastasis in the context of SCC is upheld, with many relevant studies published.

Therefore, in this separate paragraph, we will provide a simple discussion of the role of HSP27 in EMT in SCC. The original evidence demonstrating that HSP27 modulates EMT was first reported by Wei L and colleagues [68] in breast cancer. Subsequently, similar results were obtained by another independent investigation [69] carried out on prostate cancer. The two separate studies opened up a new research field, which led to a proliferation of similar studies [70,71,72] verifying the phenomenon that variations of HSP27 expression can have an effect on the EMT process, irrespective of the cancer types.

After reviewing these reports concerning HSP27 in EMT, the possible biochemical mechanism underlying HSP27-modulated EMT could be summarized as follows: first, HSP27 can promote the expression of Twist, an important transcription factor in EMT, by regulating the activated STAT3 signaling pathway. Therefore, it can be said that HSP27 and activated STAT3 are required in this phenomenon. Second, aside from Twist, another equally important transcription factor, Snail, can be bound and stabilized by HSP27, consequently inducing EMT features [72].

In contrast, the relevant studies concerning the HSP27 modulation of EMT, specifically in SCC, are superficial and scarce. Using quercetin, an inhibitor of Hsp27, Chen SF et al. [36] confirmed that the EMT process was suppressed after the inhibition of HSP27 in oral squamous cell carcinoma (OSCC) cells in vitro. However, quercetin is not a specific inhibitor of HSP27. It can block many pathways, including EMT. Therefore, care needs to be taken when interpreting the data from Chen SF et al. [36]. In addition, another study from Zhang X and coworkers [6] in our laboratory made a similar observation that SUMOylated HSP27, once genetically inhibited by transfection with short hairpin RNA (shRNA) targeted to SUMOylation, can significantly suppress EMT in ESCC, explicitly indicative of its control of EMT.

## 11. Extracellular HSP27

First discovered as part of the cellular response to increased temperature, HSP was originally found in the heat shock response by Molto MD et al. [73] and then biochemically identified by Tissieres et al. [74]. Subsequent studies revealed that HSPs correspond to a large family of proteins after a variety of stresses [75], more than just heat shock, including hyperthermia, oxygen radicals, heavy metals, ethanol and amino acid analogs. Although the most common function of HSP as a molecular chaperone is well accepted in the cytosol and other subcellular compartments, it was observed to be released into the extracellular microenvironment. The first discovery describing the presence of HSP in the extracellular environment was made for HSP70 by Tytell M et al. [76], followed by an extension to HSP71 and HSP110 by Hightower LE and colleagues [77].

Currently, most HSPs were found extracellularly, and HSP27 [78] is no exception. The original report [78] whereHSP27 was observed in the extracellular environment came from astrocytes. Extracellular HSP can be secreted by a variety of cell types and accepted by others. The roles of extracellular HSPs have little to do with their chaperone activity, which is not surprising in that extracellular HSPs are often moonlighted as signaling molecules implicated in the crosstalk among cells, inducing an array of activities.

In this scenario, the role of extracellular HSP27 is not limited to chaperones. Take HSP27 in tumor cells as an example. Thuringer D et al. [79] discovered that extracellular HSP27 mediates angiogenesis through Toll-like receptor 3 (TLR3). The HSP27/TLR3 interaction induces NF-κB activation, thereby leading to VEGF-mediated cell migration and angiogenesis in tumor cells. A good summary of the function of extracellular HSP27 as a therapeutic target in tumors was extensively reviewed elsewhere [80,81]. In contrast, studies on extracellular HSP27 in SCC are very limited.

Recently, HSP27 was found to be released into the microenvironment in the form of exosomes in ovarian cancer [82,83]. In our recent study, we isolated exosomes from peripheral blood collected from three cases of esophageal intraepithelial neoplasm (EIN), three cases of high-grade EIN, three ESCC with T2 stage and three ESCC with T3 stage; and importantly, we had the protein cargo in the exosomes identified using a proteomics approach. We found that not only HSP27 but also HSP90AA1, HSP90AB1, HSPA1A, HSP5A, HSPA8, HSP90B1 and HSPA6 were also contained within the exosomes (data unpublished), with their roles being unclear. The proteomic analyses presented here confirmed our assumption that HSP27 and its family members are key regulators that have significant effects on the pathogenesis of ESCC through their enigmatic roles. Nevertheless, the biochemical mechanism by which HSP27 is released extracellularly remains obscure and is far beyond the scope of this review. Thus, more in-depth biochemical work is necessary to demystify the problem in the future.

## 12. Inhibitors of HSP27 as a Therapeutic Strategy for Cancer

HSP27 expression is elevated in various types of SCC, including OSCC, HNSCC, TSCC, ESCC and LUSC, and it is associated with shorter survival, chemoresistance, tumor aggressiveness, and crucial tumor cell migration and invasion; thus, HSP27 has come to be a promising therapeutic target for the treatment of cancer, drawing increasing attention from the pharmaceutical industry. Considering this, it is no surprise that HSP27 can be targeted. Some specific inhibitors targeting HSP27 have been developed. For instance, OGX-427 [84], an antisense inhibitor targeting HSP27, has entered phase I clinical trials for patients with castration-resistant prostate cancer and other advanced cancers. A similar effect was observed in patients with advanced urothelial cancer treated with OGX-427 [85]. Its therapeutic value is much appreciated, and it has entered phase II evaluation using a randomized, first-line, placebo-controlled study, with preliminary data showing a promising response rate. Apatorsen [85], another kind of HSP27 inhibitor, was evaluated using a placebo-controlled, double-blind, phase II trial for patients with advanced urothelial cancer, but the results revealed that no better efficacy was observed for apatorsen combined with standard chemotherapy. The study did not demonstrate a survival benefit in the overall study population, but patients with poor prognostic features might benefit from this combination. Further exploration of apatorsen in high-risk patients is necessary. Despite advances in HSP27 inhibitor development, several challenges still exist that need to be resolved. For example, a complete understanding of the complex mechanism of HSP27 is needed, which will pave the way for the development of effective treatments. Next, the development of HSP27 inhibitors remains clinically challenging. Much more effort is therefore required to target HSP27 for the better treatment of cancers.

## 13. In Summary

At present, the pathophysiological implications of HPS27 in SCC are far from certain. Given the wide involvement of HSP27 in the tumorigenesis of SCC, the poor understanding and lack of knowledge of the underlying biochemical mechanisms are frustrating. Both the unsatisfactory understanding of the role of HSP27 in SCC and the conflicting reports require further studies with larger sample sizes and unbiased approaches. Revisiting the old data onHSP27 published earlier, we can see that the majority of studies on HSP27 usually adopted only one approach, immunohistochemistry, which heavily depends on the quality of the primary antibody and the criteria used for immunoscoring. In addition, these studies were usually performed with insufficient sample sizes, which often leads to statistical powerlessness when further stratifying the data for an analysis of the clinicopathological significance of HSP27 expression in SCC. We provided a critical review in support of the extensive engagement of HSP27, revealing its multifaceted role in the tumorigenesis of SCC.

## Figures and Tables

**Figure 1 cells-11-01665-f001:**
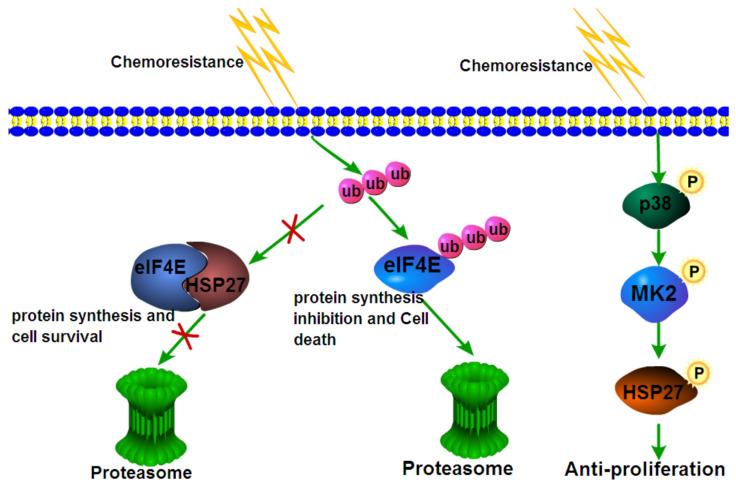
Schematic diagram summarizing the working models discovered so far regarding the biochemical mechanism by which HSP27 is involved in the chemoresistance of cancer cells on the basis of references [43,46], from which reprinted and adapted with permission was obtained. In the working model proposed by reference [43], HSP27 was found to interact with the transcription factor eIF4E, which can protect eIF4E from ubiquitination and ensuing protein degradation, thereby conferring cancer cell chemoresistance. In a working model from reference [46], activated or HSP27 phosphorylated by p38 can produce anti-proliferation effects, which renders cancer cells chemoresistant, as observed.

**Table 1 cells-11-01665-t001:** Expression patterns of HSP27 in SCC.

First Author	Type of SCC	Approach Taken	Number of Cases					Associated with Differentiation	Conclusion
			Normal Mucosa	Dysplasia	EIN	CIS	Invasive SCC		
Trautinger F et al. [2]	skin	IHC	10 all positive				6 all negative	supporting	HSP27 in the upper epidermal layers may be a marker for epidermal malignancy
Ito T et al. [14]	Tongue	IHC	Negative	positive		Positive 18/24		supporting	related to poor histological differentiation.
Lambot MA et al. [3]	Esophagus	IHC	5 all positive				21 positive	none	Hsp27 increases with the anaplasia of the ESCC.
Leonardi R et al. [15]	Esophagus	IHC	Intense staining	No or low staining			High and low staining	supporting	It is reduced in poorly differentiated areas and elevated in highly differentiated areas
Ono A et al. [16]	Cervix	IHC	Negative or low	positive	positive		positive	supporting	suggesting the role of Hsp27 in tumor development and progression.
Wang A et al. [17].	Tongue	IHC	positive	positive		positive	positive	supporting	HSP27 was associated with poor differentiation.
Tozawa-Ono A et al. [18]	Cervix	IHC	Positive 32 out of 51		Positive 44 out of 57		All positive	No	HSP27 may be a useful tool in diagnosing IN of the cervix.

Note: SCC, squamous cell carcinoma; IHC, immunohistochemistry; EIN, esophageal intraepithelial neoplasm; CIS, carcinoma in situ.

**Table 2 cells-11-01665-t002:** Clinicopathological and prognostic significance of HSP27 in SCC.

First Author	Type of SCC Used	Approach Taken	Stage	Grade	Metastasis	Therapeutic Effect	Prognosis	Conclusion
Gandour-Edwards R et al. [24]	Head and neck	IHC	Non associated	Non associated	Non associated	Not mentioned	Non	Remains unclear.
Ito T et al. [14]	Tongue	IHC	Non	Non	Non	Not mentioned	Non	Remains unclear.
Shiozaki H et al. [25]	Esophagus	IHC	Non	Non	Non	Not mentioned	associated	Associated with poor survival.
Takeno S et al. [26]	Esophagus	IHC	Non	Non	Non	Associated	Non	Hsp27 can predict the therapeutic effect.
Mese H et al. [19].	Esophagus	IHC	Non	Non	Not mentioned	Not mentioned	associated	Independent prognostic factor.
Lo Muzio L et al. [11].	Oral	IHC	associated	associated	Not mentioned	Not mentioned	associated	Could be a novel diagnostic and prognostic factor.
Miyazaki T et al. [27]	Esophagus	IHC	Non	Non	Non	Most reliable predictor	Non	Hsp27 was the most reliable predictor for chemo-or radiotherapy.
Lo Muzio L et al. [12]	Head and neck	IHC	Non	Non	Non		associated	Reduced expression of HSP27 is an early marker of poor prognosis.
Wang A et al. [17]	Tongue	IHC	Non	Non	Non	Not mentioned	Associated	Hsp27 appears to be an independent prognostic marker.
Luz CC et al. [22]	Esophagus	IHC	Non	Non	Non	Not mentioned	Non	HSP27 was not a good prognostic factor for ESCC.
Zhang Y et al. [23]	Esophagus	IHC	Non	Non	associated	Not mentioned	Non	HSP27 could be used as a prognostic factor in ESCC.
Ajalyakeen H et al. [28]	Oral	IHC	Not mentioned	associated	Not mentioned	Not mentioned	Not mentioned	HSP27 may be an indicator of the biological behavior of the tumor.

## Data Availability

Not applicable.

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
