# Peer review of "Revisiting the Old Data of Heat Shock Protein 27 Expression in Squamous Cell Carcinoma: Enigmatic HSP27, More Than Heat Shock"

_cells, 2022, doi:10.3390/cells11101665_

Round 1
Reviewer 1 Report
The paper requires dramatic improvement of English language. I cannot review it before the improvements are made.
Author Response
This is an interesting, comprehensive review dedicated to the important problem, namely a role of HSP27 in squamous cell carcinoma. The submitted material may be helpful for readers of Cells, especially for molecular oncologists. I have only minor remarks on improvement of the manuscript:
Response: we appreciate the reviewer so much for the comments given to our paper that” This is an interesting, comprehensive review dedicated to the important problem, namely a role of HSP27 in squamous cell carcinoma. The submitted material may be helpful for readers of Cells, especially for molecular oncologists.”
The Authors should add information about implication of HSP27 in the cytoprotection from oxidative stress. This HSP27-mediated mechanism is thought to contribute to chemo- and radioresistance of cancer cells (similar mechanisms may take place in squamous cell carcinoma);
Response: yep, this is pertinent suggestion that we appreciate. We have included information regarding implication of HSP27 in the cytoprotection from oxidative stress at the suggestion of you, which was separately paragraphed with the subtitle that protection against oxidative stress.
Protection against oxidative stress
Apart from engagement in chemo- and radioresistance, HSP27 has also been discovered to take part in cytoprotection from apoptosis induced by oxidative stress[51]. In regard to protection from oxidative stress, there remains some disagreement over whether it is phosphorylated or unphosphorylated HSP27 that plays a protective role against cell death and apoptosis mainly caused by oxidative stress, with some [52, 53] supporting that HSP27 exerts a protective role, whereas others [54, 55] argue that only phosphorylated HSP27 protects against oxidative stress.
Accumulating evidence [56] has shown that phosphorylation of HSP27 can be stimulated by a variety of stresses in addition to heat shock or hyperthermia, for example, oxidative stress or exposure to cytotoxic drugs (heavy metals, chemotherapy, alcohol). It can be imagined that when in such an environment, cells have to establish a protective mechanism to survive the adverse conditions and maintain their functions. Oxidative stress, generally referred to as excess production of reactive oxygen species (ROS) relative to antioxidant defense, can cause large amounts of ROS formation within cells, which in turn further amplifies oxidative stress, eventually leading to cell death and apoptosis. A great deal of experimental data has indicated that HSP27 has a protective effect against oxidative stress owing to its ability to modulate intracellular redox homeostasis. Intracellular large aggregates of HSP27 have been shown to be capable of regulating ROS production and therefore generating cytoprotection against oxidative stress [57]. Nevertheless, this activity can be somewhat mitigated by phosphorylation of HSP27, mainly through dissociating the HSP27 complexes into tetramers [58]. Consistently, one study contributed by Huot J et al. [59] revealed that overexpression of the wild-type, as opposed to nonphosphorylated HSP27 from humans, led to increased resistance against oxidative stress and enhanced survival after treatment with H2O2. Basically, in line with this report, another study [60] held a similar opinion that phosphorylated HSP27 seems to be dispensable for the protection of HSP27 against H2O2-induced oxidative stress. These data strongly suggest that HSP27 exerts cytoprotection against oxidative stress via different mechanisms, such as interactions with client proteins [61], prevention of protein aggregation and protection of the cytoskeleton. Considering the close relationship [62-64] between the development of SCC and oxidative stress, the studies mentioned above may help account for the phenomenon that high expression of HSP27 is frequently present in tumorous tissues relative to paired normal controls.
The reference newly included:
- Arrigo A. P.;Firdaus W. J.;Mellier G.;Moulin M.;Paul C.;Diaz-latoud C. and Kretz-remy C. Cytotoxic effects induced by oxidative stress in cultured mammalian cells and protection provided by Hsp27 expression. Methods 2005, 35:126-138.doi: 10.1016/j.ymeth.2004.08.003
- Zhang H. L.;Jia K. Y.;Sun D. and Yang M. Protective effect of HSP27 in atherosclerosis and coronary heart disease by inhibiting reactive oxygen species. J Cell Biochem 2019, 120:2859-2868.doi: 10.1002/jcb.26575
- Ramani S. and Park S. HSP27 role in cardioprotection by modulating chemotherapeutic doxorubicin-induced cell death. J Mol Med (Berl) 2021, 99:771-784.doi: 10.1007/s00109-021-02048-4
- Shimada Y.;Tanaka R.;Shimura H.;Yamashiro K.;Urabe T. and Hattori N. Phosphorylation enhances recombinant HSP27 neuroprotection against focal cerebral ischemia in mice. Neuroscience 2014, 278:113-121.doi: 10.1016/j.neuroscience.2014.07.073
- Gaitanaki C.;Konstantina S.;Chrysa S. and Beis I. Oxidative stress stimulates multiple MAPK signalling pathways and phosphorylation of the small HSP27 in the perfused amphibian heart. J Exp Biol 2003, 206:2759-2769.doi: 10.1242/jeb.00483
- Mymrikov E. V.;Seit-Nebi A. S. and Gusev N. B. Large potentials of small heat shock proteins. Physiol Rev 2011, 91:1123-1159.doi: 10.1152/physrev.00023.2010
- Mehlen P.;Hickey E.;Weber L. A. and Arrigo A. P. Large unphosphorylated aggregates as the active form of hsp27 which controls intracellular reactive oxygen species and glutathione levels and generates a protection against TNFalpha in NIH-3T3-ras cells. Biochem Biophys Res Commun 1997, 241:187-192.doi: 10.1006/bbrc.1997.7635
- Rogalla T.;Ehrnsperger M.;Preville X.;Kotlyarov A.;Lutsch G.;Ducasse C.;Paul C.;Wieske M.;Arrigo A. P.;Buchner J. and Gaestel M. Regulation of Hsp27 oligomerization, chaperone function, and protective activity against oxidative stress/tumor necrosis factor alpha by phosphorylation. J Biol Chem 1999, 274:18947-18956.doi: 10.1074/jbc.274.27.18947
- Huot J.;Houle F.;Spitz D. R. and Landry J. HSP27 phosphorylation-mediated resistance against actin fragmentation and cell death induced by oxidative stress. Cancer Res 1996, 56:273-279
- Preville X.;Gaestel M. and Arrigo A. P. Phosphorylation is not essential for protection of L929 cells by Hsp25 against H2O2-mediated disruption actin cytoskeleton, a protection which appears related to the redox change mediated by Hsp25. Cell Stress Chaperones 1998, 3:177-187.doi: 10.1379/1466-1268(1998)003<0177:pinefp>2.3.co;2
- Davila D.;Jimenez-Mateos E. M.;Mooney C. M.;Velasco G.;Henshall D. C. and Prehn J. H. Hsp27 binding to the 3'UTR of bim mRNA prevents neuronal death during oxidative stress-induced injury: a novel cytoprotective mechanism. Mol Biol Cell 2014, 25:3413-3423.doi: 10.1091/mbc.E13-08-0495
- Beevi S. S.;Rasheed M. H. and Geetha A. Evidence of oxidative and nitrosative stress in patients with cervical squamous cell carcinoma. Clin Chim Acta 2007, 375:119-123.doi: 10.1016/j.cca.2006.06.028
- Bentz B. Head and neck squamous cell carcinoma as a model of oxidative-stress and cancer. J Surg Oncol 2007, 96:190-191.doi: 10.1002/jso.20817
- Salzman R.;Pacal L.;Kankova K.;Tomandl J.;Horakova Z.;Tothova E. and Kostrica R. High perioperative level of oxidative stress as a prognostic tool for identifying patients with a high risk of recurrence of head and neck squamous cell carcinoma. Int J Clin Oncol 2010, 15:565-570.doi: 10.1007/s10147-010-0108-z
2.It would be nice, if the Authors discuss potential ways and approaches to targeting HSP27 in order to treat squamous cell carcinoma. I mean small molecule inhibitors of HSP27, natural compounds, HSP27-targeting nanoparticles etc that demonstrated promising results in models of anticancer therapy.
Response: fully agreeing with the reviewer, we appreciate the reviewer for the constructive advice. As suggested, we have made some discussion surrounding inhibitors of HSP27, which was also paragraphed separately with the subtitle that Inhibitors of HSP27 as therapeutic strategy of cancer. In order to help your check, the separate paragraph was also posted below:
Inhibitors of HSP27 as therapeutic strategy of cancer
HPS7 expression is elevated in various types of SCC, including OSCC, HNSCC, TSCC, ESCC and LUSC, and it is associated with shorter survival, chemoresistance, tumor aggressiveness, and crucial tumor cell migration and invasion, and thus HSP27 has come to be a promising therapeutic target for the treatment of cancer, drawing increasing attention from the pharmaceutical industry. Considering this, it is no surprise that HSP27 can be targeted. Some specific inhibitors targeting HSP27 have been developed. For instance, OGX-427 [84], an antisense inhibitor targeting HSP27, has entered phase I clinical trials for patients with castration-resistant prostate cancer and other advanced cancers. A similar effect was observed in patients with advanced urothelial cancer treated with OGX-427 [85]. Its therapeutic value has been much appreciated and it has entered phase II evaluation using a randomized, first-line, placebo-controlled study, with preliminary data showing a promising response rate. Apatorsen [85], another kind of HSP27 inhibitor, was evaluated using a placebo-controlled, double-blind, phase II trial for patients with advanced urothelial cancer, but the results revealed that no better efficacy was observed for apatorsen combined with standard chemotherapy. The study did not demonstrate a survival benefit in the overall study population, but patients with poor prognostic features might benefit from this combination. Further exploration of apatorsen in high-risk patients is necessary. Despite advances in HSP27 inhibitor development, several challenges still exist that need to be resolved. For example, a complete understanding of the complex mechanism of HSP27 is needed, which will pave the way for the development of effective treatments. Next, the development of HSP27 inhibitors remains clinically challenging. Much more effort is therefore required to target HSP27 for better treatment of cancers.
The reference newly included:
- Chi K. N.;Yu E. Y.;Jacobs C.;Bazov J.;Kollmannsberger C.;Higano C. S.;Mukherjee S. D.;Gleave M. E.;Stewart P. S. and Hotte S. J. A phase I dose-escalation study of apatorsen (OGX-427), an antisense inhibitor targeting heat shock protein 27 (Hsp27), in patients with castration-resistant prostate cancer and other advanced cancers. Ann Oncol 2016, 27:1116-1122.doi: 10.1093/annonc/mdw068
- Bellmunt J.;Eigl B. J.;Senkus E.;Loriot Y.;Twardowski P.;Castellano D.;Blais N.;Sridhar S. S.;Sternberg C. N.;Retz M.;Pal S.;Blumenstein B.;Jacobs C.;Stewart P. S. and Petrylak D. P. Borealis-1: a randomized, first-line, placebo-controlled, phase II study evaluating apatorsen and chemotherapy for patients with advanced urothelial cancer. Ann Oncol 2017, 28:2481-2488.doi: 10.1093/annonc/mdx400

Reviewer 2 Report
Title: Revisiting the old data of Heat shock protein 27 expression in squamous cell carcinoma: enigmatic HSP27, more than heat shock
Authors: Shutao Zheng , Yan Liang , Lu Li , Yiyi Tan , Qing Liu , Tao Liu , Xiaomei Liu
REVIEWER'S COMMENTS:
This is an interesting, comprehensive review dedicated to the important problem, namely a role of HSP27 in squamous cell carcinoma. The submitted material may be helpful for readers of Cells, especially for molecular oncologists. I have only minor remarks on improvement of the manuscript:
1.The Authors should add information about implication of HSP27 in the cytoprotection from oxidative stress. This HSP27-mediated mechanism is thought to contribute to chemo- and radioresistance of cancer cells (similar mechanisms may take place in squamous cell carcinoma);
2.It would be nice, if the Authors discuss potential ways and approaches to targeting HSP27 in order to treat squamous cell carcinoma. I mean small molecule inhibitors of HSP27, natural compounds, HSP27-targeting nanoparticles etc that demonstrated promising results in models of anticancer therapy.
Author Response
Comments and Suggestions for Authors
The paper requires dramatic improvement of English language. I cannot review it before the improvements are made.
Response: we have the paper improved in terms of English language by professional service from AJE (short for American Journal of Experts). The credential of language polishing was posted below for your check.

This manuscript is a resubmission of an earlier submission. The following is a list of the peer review reports and author responses from that submission.
Round 1
Reviewer 1 Report
Shutao Zheng and co-authors' submitted a Manuscript that comprehensively reviews the role of HSP27 in squamous cell carcinoma. The authors analyzed a large number of papers published on the theme. This review will be helpful in the re-evaluation of the role of HSP27 and will be interesting for the readers of Cells from the cancer research field.
The main drawback of the Manuscript is the author's decision to narrow themselves to reading papers about HSP27 in squamous cell carcinoma only. This decision led to inconclusive statements such as "Despite several mechanisms by which HSP27 engaged in chemoresistance have been put forward, attempting to explain in these studies; these were not fully elucidated…" The general mechanisms linking HSP27 to chemoresistance are very well studied and described in many papers for many other cancer types. Squamous cell carcinoma cells use similar mechanisms of chemoresistance, or the author would like to propose a specific mechanism for this type of cancer. These general mechanisms should be discussed or presented as a figure in the Manuscript.
The abbreviations HNSCC, ESCC, are not annotated in the Manuscript.
"Using Quercetin, an inhibitor of Hsp27"- quercetin is not a specific inhibitor of HSP27. It can trigger many pathways. The authors should mention this fact in the Manuscript.
There are repetitive sub-paragraphs in the text: " 1.5. Involvement in metastases" and "1.8. Taking part in epithelial-mesenchymal transition (EMT) and metastasis". It is better to join them in one.
The native speaker should check the English language of the Manuscript. There are many sentences in the Manuscript which are hard to read.
"Despite several mechanisms by which HSP27 engaged in chemoresistance have been put forward, attempting to explain in these studies; these were not fully elucidated, with association or correlational relationship being investigated between up-regulated HSP27 and chemoresistance."
There are many minor errors: "Nowadays, all most all HSPs" – almost all.
Author Response
Comments and Suggestions for Authors
Shutao Zheng and co-authors' submitted a Manuscript that comprehensively reviews the role of HSP27 in squamous cell carcinoma. The authors analyzed a large number of papers published on the theme. This review will be helpful in the re-evaluation of the role of HSP27 and will be interesting for the readers of Cells from the cancer research field.
The main drawback of the Manuscript is the author's decision to narrow themselves to reading papers about HSP27 in squamous cell carcinoma only. This decision led to inconclusive statements such as "Despite several mechanisms by which HSP27 engaged in chemoresistance have been put forward, attempting to explain in these studies; these were not fully elucidated…" The general mechanisms linking HSP27 to chemoresistance are very well studied and described in many papers for many other cancer types. Squamous cell carcinoma cells use similar mechanisms of chemoresistance, or the author would like to propose a specific mechanism for this type of cancer. These general mechanisms should be discussed or presented as a figure in the Manuscript.
Response: thank you for your feedback and critical critique on our review titled “Revisiting the old data of Heat shock protein 27 expression in squamous cell carcinoma: enigmatic HSP27, more than heat shock”, which we found very useful. As suggested, we have rephrased some sentences, especially those pointed out by the reviewer and some else the reviewer had not explicitly indicated were also recast.
With all due respect, we see it differently when it comes to the contention of the reviewer that “the author's decision to narrow themselves to reading papers about HSP27 in squamous cell carcinoma only”. Compared with adenocarcinoma, squamous cell carcinoma (SCC), actually, were clinically and generally regarded to be refractory to the chemo-and-radio therapy. Considering this, in our viewpoint, it would be more meaningful to have a review surrounding involvement of HSP27 in SCC, rather than expanding on adenocarcinoma. On the flip side, SCC, in particular, esophageal squamous cell carcinoma (ESCC) and cervical squamous cell carcinoma (CSCC), two common SCC, have been highly prevalent in Xinjiang area where we have been living and working. So, paying attention to SCC would have more practical significance. Out of thoughts mentioned above, we were determined to narrow down all the relevant literatures regarding HSP27 to just focus on SCC.
Given the chemoresistance mediated by HSP27 in the setting of SCC, which has been enigmatic and obscure till now, we have included some words summarizing the latest advance published resolving around HSP27 in SCC. However, the literatures regarding engagement of HSP27 in the chemoresistance of SCC have been not much, even scarce. Despite the engagement of HSP27 in chemoresistance of SCC has been determined in outline, however, much details of working mechanism were missing. In light of this, it would be hard to provide a schematic diagram favoring understanding how did HSP27 work in the chemo-or-radio resistance of SCC, as suggested. We hope the reviewer could understand.
The abbreviations HNSCC, ESCC, are not annotated in the Manuscript.
Response: helpful tips that we appreciate so much. All of these contractions have been spelled out when first appeared and have them annotated where appropriate in the main text at the suggestion given.
"Using Quercetin, an inhibitor of Hsp27"- quercetin is not a specific inhibitor of HSP27. It can trigger many pathways. The authors should mention this fact in the Manuscript.
Response: excellent question that we appreciate! Agreeing with the reviewer, we have mentioned the fact—quercetin is not a specific inhibitor of HSP27 in the main text where appropriate, at the request of the reviewer.
There are repetitive sub-paragraphs in the text: " 1.5. Involvement in metastases" and "1.8. Taking part in epithelial-mesenchymal transition (EMT) and metastasis". It is better to join them in one.
Response: with all due respect, we respectfully disagreed with the reviewer mainly out of the two reasons listed below: 1. As stated in the main text, the conventional wisdom held that EMT was closely related to metastasis of cancer; however, this deep-seated notion has been challenged by two pieces of seminal articles published in journal Nature demonstrating that EMT was dispensable for metastasis but more likely involved in chemoresistance of cancer. Given this controversial point, in our viewpoint, subsection 1.8 Taking part in epithelial-mesenchymal transition (EMT) should be written as a separate paragraph, rather than merged with the subsection 1.5 Involvement in metastases as suggested. 2. To avoid the potential confusing that may be caused by the subtitle “1.8. Taking part in epithelial-mesenchymal transition (EMT) and metastasis”, we have it rephrased as 1.8. Taking part in epithelial-mesenchymal transition (EMT)”, that is to say, we have the word “metastasis” deleted in subtitle.
The native speaker should check the English language of the Manuscript. There are many sentences in the Manuscript which are hard to read.
Response: That’s understandable! As investigator of English as a Second Language (ESL), language problem when writing is inevitable. However, we have tried to weed out the entire language problem as much as we can. It is hopeful that all that we did would be OK.
"Despite several mechanisms by which HSP27 engaged in chemoresistance have been put forward, attempting to explain in these studies; these were not fully elucidated, with association or correlational relationship being investigated between up-regulated HSP27 and chemoresistance." There are many minor errors: "Nowadays, all most all HSPs" – almost all.
Response: we have weeded out these problems relating to language writing and won’t let these slip into our main text as much as we can.

Reviewer 2 Report
This is a nice review about the role of HSP27 in the development of cell carcinoma. I have just a few minor concerns that should be addressed by the reviewers before acceptance for publication.
First of all, I think that the authors are missing to discuss a very important issue in malignant cells proliferation, i.e. the crosstalk between HSP27 and the proteasome. Indeed, it has been observed that y activating p38 mitogen activated protein kinase, the proteasomal inhibitor MG-132 increases Hsp27 phosphorylation and the production of tiny oligomers in oligodendroglial cells. After 24 hours of treatment, cell lines had generated protein accumulations near the centrosome, however aggresome formation was delayed in cells overexpressing Hsp27 compared to cells that did not express Hsp27. (see: e.g. J. Neurochem. (2010) 114, 960–971). For a comprehensive review on the role of proteasome in cancer and cell proliferation the authors could refer to: (Pharmacology & Therapeutics 213 (2020) 107579. I suggest including this important issue in this otherwise complete review.
Author Response
Comments and Suggestions for Authors
This is a nice review about the role of HSP27 in the development of cell carcinoma. I have just a few minor concerns that should be addressed by the reviewers before acceptance for publication.
First of all, I think that the authors are missing to discuss a very important issue in malignant cells proliferation, i.e. the crosstalk between HSP27 and the proteasome. Indeed, it has been observed that y activating p38 mitogen activated protein kinase, the proteasomal inhibitor MG-132 increases Hsp27 phosphorylation and the production of tiny oligomers in oligodendroglial cells. After 24 hours of treatment, cell lines had generated protein accumulations near the centrosome, however aggresome formation was delayed in cells overexpressing Hsp27 compared to cells that did not express Hsp27. (see: e.g. J. Neurochem. (2010) 114, 960–971). For a comprehensive review on the role of proteasome in cancer and cell proliferation the authors could refer to: (Pharmacology & Therapeutics 213 (2020) 107579. I suggest including this important issue in this otherwise complete review.
Response: yeah, this is an important and constructive issue that would help strengthen our review that we appreciate. However, after closer examination of these references mentioned by the reviewer, we found that the literatures, despite closely related to HSP27, in effect had little to do with the our main theme framed from the outset, that is to say, engagement of HSP27 in SCC. Inciting these references and ensuing discussions made about those will inevitably lead to the dilution and digression of our main purpose. Given this, these literatures would undoubtedly lend little support to our building the review. Anyway, to meet the demands of the review, we have made some discussion surrounding the implication of HSP27 in proliferation of SCC cells, doing our best. Hope all that we did can make you satisfied.
